# Remodeling and self-healing of individual amyloid tactoids via multiphoton absorption

Dongdong Lin [1,2,3] ✉, Hamed Almohammadi [1,4], Yufen Zhao [5] &
Raffaele Mezzenga [1,6] ✉

Colloidal self-assembly can typically be controlled only globally, at $\geq 10^1$ μm scale, such as in liquid-liquid crystalline phase separation (LLCPS) of anisotropic tactoidal droplets. Here, we introduce a stimuli-responsive approach allowing a fully reversible structural control of LLCPS morphologies at the individual droplet level. Using amyloid-based tactoids and multiphoton absorption with localized photothermal effect, we can cut, ablate, and self-heal individual tactoids, and -when desired- print, erase, and store structural information within at sub-micron resolution. We exploit the nematic-isotropic-nematic transition within single tactoids to locally melt the liquid crystalline (LC) order into the isotropic phase, leaving unchanged the bulk dispersion structure. The locally melted nematic field recovers its ground-state LC order within minutes after the exposure, showing both self-healing and short-term memory-storage features. Furthermore, hybrid cholesteric tactoids functionalized by guest nanoparticles can be re-modeled into new tactoids with different symmetry and functionalities, for example, featuring laser-induced fluorescent encoding. These results introduce a general strategy to direct phase separation-within-phase separation and to store, control, and engineer information at the sub-micron level in heterogeneous complex fluids.

Liquid crystalline (LC) droplets, also referred to as tactoids, spontaneously nucleate and grow within an isotropic solution, forming a heterogeneous water-in-water system in which the dispersed phase emerges primarily (but not exclusively) as a trade-off between orientational and free volume packing entropies[1–5]. These droplets hold various spherical, ellipsoidal, and spindle-shaped morphologies with intricate internal symmetries and physical properties[6–8]. Tactoids have been observed in a variety of LC systems, including polypeptides[9,10], supramolecules[6], molecular aggregates[11], biological colloids of viruses[12,13], nanocrystals[14], and fibrils[8,15,16]. Significant efforts have been devoted to understanding the tactoids' origin, shape, and structure, which may not only enable novel applications[5,17,18], but also enhance our understanding of a number of biological phenomena[19], including the formation of collagen byssal threads[20], amyloid protein condensates and plaques[21,22], and P. aeruginosa resistance to antibiotics[13,23]. Numerous studies have explored controlling the pathway leading to the formation of tactoids emerging from liquid–liquid crystalline phase separation (LLCPS)[5,24,25], using intensive variables, such as temperature[26,27], pressure[28], ion type[29], concentration[25,30], pH[31], and other additives[6,9,16], which therefore affect the entire system. At best, tuning the morphology of a restricted number of tactoids has been achieved through microfluidics approaches[15,32,33]. However, manipulation of the internal structure, shape, and physical properties at the single tactoid level is yet to be reported. Because this can only be attempted by departing from the thermodynamic ground state, non-invasive localized stimuli have to be considered to pursue this task.

[1]Department of Health Sciences & Technology, ETH Zurich, Zurich, Switzerland. [2]Institute of Fundamental Physics and Quantum Technology, Ningbo University, Ningbo, Zhejiang, PR China. [3]School of Physical Science and Technology, Ningbo University, Ningbo, PR China. [4]John A. Paulson School of Engineering and Applied Sciences, Harvard University, Cambridge, MA, USA. [5]Qian Xuesen Collaborative Research Center of Astrochemistry and Space Life Sciences, Ningbo University, Ningbo, PR China. [6]Department of Materials, ETH Zurich, Zurich, Switzerland. ✉e-mail: lindongdong@nbu.edu.cn; raffaele.mezzenga@hest.ethz.ch

Unlike the linear absorption of light by matter, multiphoton absorption (MPA) is a second and third-order nonlinear effect, in which the absorption is greatly strengthened as energy density increases[34–36]. Previous research confirmed that highly ordered amyloid fibrils and silk fibroin exhibit strong nonlinear optical absorption with large multiphoton cross-section, which is absent in their native protein precursors[37,38]. Infrared pulses with low average energy exhibit high penetration in biological colloidal media and are absorbed by amyloids in multiphoton mode[39,40]. This property allows minimal surface absorption (in accordance with the Lambert–Beer law) and enables the formation of a focal spot, causing MPA deep within the materials. Multiphoton-induced structural modification and patterning have been investigated in various biocompatible materials, including silk fibroin[36], collagen[41], poly(vinyl-alcohol)[42], poly(methyl methacrylate)[43], and gelatin hydrogels[44], but to the best of our knowledge, this technique has not yet been considered as a tool to control LLCPS.

In this article, we report a method to manipulate amyloid LC droplets via nonlinear MPA in three dimensions (3D) with sub-micron resolution (named 3D-Heating). Amyloid tactoids show appealing capability of being locally heated through nonlinear absorption, in situ, with a high degree of spatial selectivity. The tactoids could be easily erased, encoded with structural information, and modulated into new tactoids by 3D-Heating. We demonstrate that the amyloid tactoids exhibit a strong memory effect, pitch-tunable function, and fast self-healing ability, thereby underscoring their potential as smart biomaterials and bio-devices through LLCPS. The resultant 3D-Heating on amyloid tactoids offers a promising strategy in the manipulation of complex fluids, with potential applications in engineering, printing, memory-storage, and self-healing in nanotechnology, life sciences, and biomaterials design.

## Results

We realized LLCPS and the formation of various tactoid symmetries using amyloid fibrils made of β-lactoglobulin (BLG) (see Fig. 1a), a readily available protein associated with dairy byproducts. Regarding pathological, functional, and artificial amyloid fibrils, their highly ordered and tightly packed core with β-sheet structures endows BLG amyloid fibrils with high mechanical rigidity and thermodynamic stability. To form amyloid tactoids, the long semi-flexible BLG amyloid fibrils with a persistence length $Lp$ of 1.8 μm were broken down by stirring into shorter fibrils (Fig. 1b and Supplementary Fig. 1). The average fibril length, measured via atomic force microscopy (AFM) images using FiberApp contour tracking software[45], is $L$ = 435 nm and the average height ~ 4 nm, corresponding to an aspect ratio of ~ 109. Next, an aqueous suspension of shortened amyloid fibrils was prepared at a concentration such that it falls within the isotropic-nematic two-phase region (Supplementary Fig. 2). At such concentration, when sealed in a capillary tube, the amyloid fibril suspension undergoes LLCPS, forming, over the course of the following weeks, tactoids with a rich palette of symmetries, sizes, and morphologies (Fig. 1b and Supplementary Fig. 3).

### Multiphoton absorption of tactoids

We discovered that a pulsed infrared laser could destruct the tactoids, whereas a laser in the visible light region did not, while performing fluorescence imaging of amyloid tactoids. Inspired by this, we set out to systematically study the laser-induced manipulation of amyloid tactoids, using the layer-by-layer exposure concepts from 3D printing technology[46–48]. We exposed individual tactoids to a scanning focused pulsed laser (810 nm, ~100 fs, 80 MHz ±1 MHz) following spatially predefined paths (Fig. 1c). During laser pulse exposure, the virtual-state lifetime is determined by the pulse duration. Intuitively, the transition of an electron from the ground state to the virtual state, and finally to the excited state, is induced by two or more sequential photons. Using z-scan techniques, we found that BLG fibrils exhibited

three-photon absorption (3PA) capabilities (Fig. 1d and the theory in Supplementary Note 1). The linear absorption spectrum showed a single absorption peak at ~280 nm from the amyloid fibrillar suspension (Fig. 1e), providing further evidence of the nonlinear absorption process under pulsed laser. Previously, the 3PA process was observed in silk-fibroin[36], amyloid fibrils[37], and spider silk[49]. The enhanced 3PA in amyloid fibrils is attributed to through-space dipolar coupling between the excited states of aromatic amino acids densely packed within the fibrous structures[37]. This highly ordered β-sheet structure was confirmed by Thioflavin T fluorescence and Fourier transform infrared spectroscopy (Supplementary Fig. 4). The heating accumulation from direct 3PA allowed us to raise the temperature of amyloid mesogens precisely at the focused point. We found that the local transition from the anisotropic to isotropic phase in tactoids required only a weak laser power of ~0.25 nJ per pulse, which was one order of magnitude lower than the energy needed for patterning in soft silk protein hydrogels (2 nJ per pulse to break hydrogels)[36].

Systematically, the relationship between the laser-induced nematic-isotropic transition rate and laser power was investigated. As shown in Fig.1f, we found that a power of 13.6 mW (total pixel dwell time of 16 μs) could have distinct breaking effect on cholesteric structures. Here, to show the relation between the irradiation area and the actual area affected within the sample, we designed three different irradiation areas (48.6, 176.4 and 360.6 μm²) as shown in Supplementary Fig. 5 and irradiated them under the same conditions. Following the irradiation, we immediately measured the disordered regions within the sample, marked with red dash circles in Supplementary Fig. 5b. Our measurements show that the area set by the program and the disordered region measured within the samples are of comparable size (Supplementary Fig. 5c), indicating the same irradiation effects on different areas. We note that the slightly lower values for disordered areas compared to the set region are due to a measurement delay (> 10 s) between the switch of NIR irradiation and POM imaging. Additionally, the kinetic irradiation was also measured in Supplementary Fig. 5d. The result shows that the isotropic region (121.9 μm²) obtained by ON-OFF laser period (pixel dwell time of 0.4 μs with three repetitions) was smaller than the single continuous exposure (167.1 μm², pixel dwell time of 1.2 μs). Over all, this method provided us with a powerful and versatile method for generating complex 3D disordered phase patterns within the otherwise LC-ordered state of the tactoid.

### Erasing and cutting amyloid tactoids

First, a bipolar tactoid, in which the director field follows the tactoid interface, was selected and irradiated entirely by exposing a cuboid volume at its center (Fig. 2a). Monitoring the morphology over time under polarized optical microscopy (POM), we observed that the tactoid with an initial volume of 17,236 μm³ disappeared within ~155 s, following an initial lag phase, indicating an accumulation of light energy before the order was erased (for volume calculation see Supplementary Fig. 6 and Supplementary Note 2). The aspect ratio ($H/2R$) of the exposed tactoid exhibited an initial decrease followed by an increase after reaching a critical point (80 s, 4,836 μm³). To explore this phenomenon, we compared the dynamics of $H$ and $2R$ during this exposure (see POM intensity maps in Supplementary Fig. 7). The length of the $H$ dimension was reduced on both poles of the bipolar, while $2R$ decreased slowly until ~80 s. We attribute this phenomenon to the different fibrillar alignments along $H$ and $2R$, leading to different ablating and expanding features. In contrast, when only half of the bipolar tactoid was exposed (Fig. 2c), the aspect ratio showed a linear increase. Remarkably, the bipolar volume decreased with time while the tactoid always maintained a spindle shape, and finally reached zero. These findings strongly suggest a rapid dynamic reconstruction of the amyloid tactoids under localized irradiation, similar to proteins in coacervates, which remain very dynamic on sub-microsecond

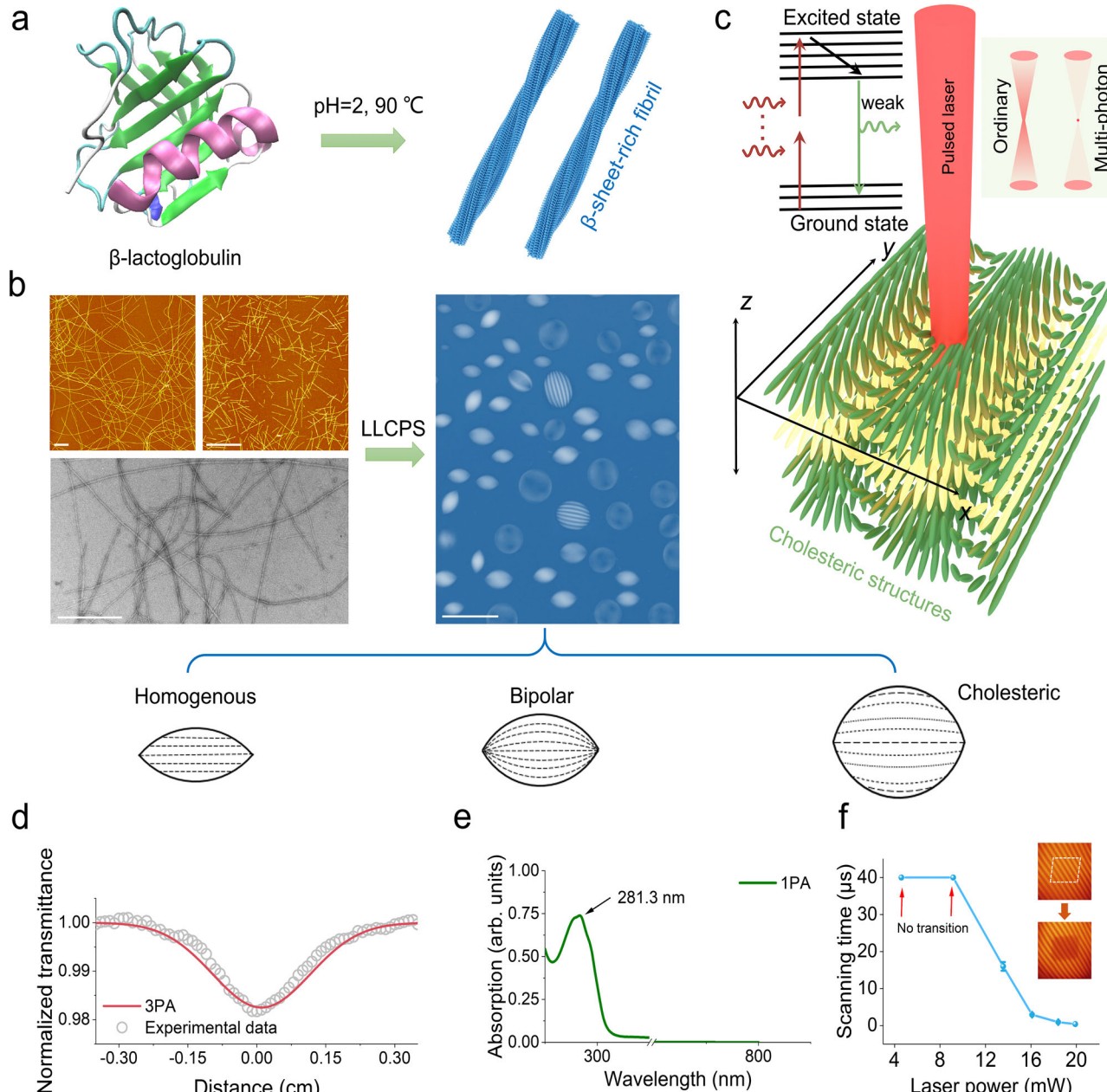

**Fig. 1 | The LLCPS of β-lactoglobulin (BLG) amyloid fibrils and the multiphoton adsorption of amyloid tactoids. a.** Amyloid fibrils with β-sheet-rich structures were obtained from the incubation of β-lactoglobulin (BLG) protein at pH=2 and 90 °C conditions. Protein structure is available from the Protein Data Bank (PDB ID: 3BLG). **b.** The AFM images of amyloid fibrils before (left panel: top-left) and after (left panel: top-right) being shortened into uniform length, and the TEM image (left panel: bottom) of the original amyloid fibrils. The scale bar is 500 nm. When prepared at a concentration within the Onsager binodal lines, the suspension of short amyloid fibrils shows the formation of a rich variety of tactoids through LLCPS (right panel, image is taken between crossed polarizers). The schematic shows the director field orientation for three different classes of homogenous, bipolar, and cholesteric tactoids. The scale bar is 200 μm. **c.** Schematic showing the exposure of the LC cholesteric structure to the pulsed laser. Compared to the ordinary laser focus, the multiphoton absorption can trigger the heating precisely on a small

focused spot. This spot can be scanned spatially in *xyz*-axes. **d.** Open aperture *z*-scan transmission curve from BLG fibrils solution with theoretical fitting for three-photon absorption (3PA). The sample was exposed to 100 fs pulses at 810 nm wavelength with a pulse repetition rate of 80 MHz. **e.** Linear absorption spectrum of BLG fibrils with background subtracted at one-photon absorption (1PA) mode. The absorption was measured in the range between 200 and 800 nm. **f.** The relationship between the scanning time (pixel dwell time) of the laser-induced nematic-isotropic transition and the laser power. The scanning time was calculated following a single pixel dwell time multiplied by the number of repetitions. Here, a two-dimensional single scanning layer inside of a cholesteric tactoid was exposed under different laser powers. The dark region represents the localized disordered state of amyloid fibrils due to the laser exposure. The data were expressed as mean ± standard deviation (S.D). Panels **a** and **c** were created with PyMOL 3.1 and 3ds Max 2021.

timescales[50]. The same protocol was applied to cholesteric tactoid (with an oblate shape, and thus with an initial aspect ratio <1) with chiral alignment of the mesogens (Fig. 2e). The 3D-Heating technique ablated the entire cholesteric tactoid in 1,718 s, despite its volume being 42 times larger than the bipolar tactoid in Fig. 2a. In this case, the

system revealed a clear lag phase in the initial ~350 s (Fig. 2f). When only half of the cholesteric tactoid was irradiated, the unexposed part retained its original chiral symmetry, but underwent internal dynamic changes into a new cholesteric tactoid (Fig. 2g). By examining the unexposed part in more detail, it was found that the half pitch

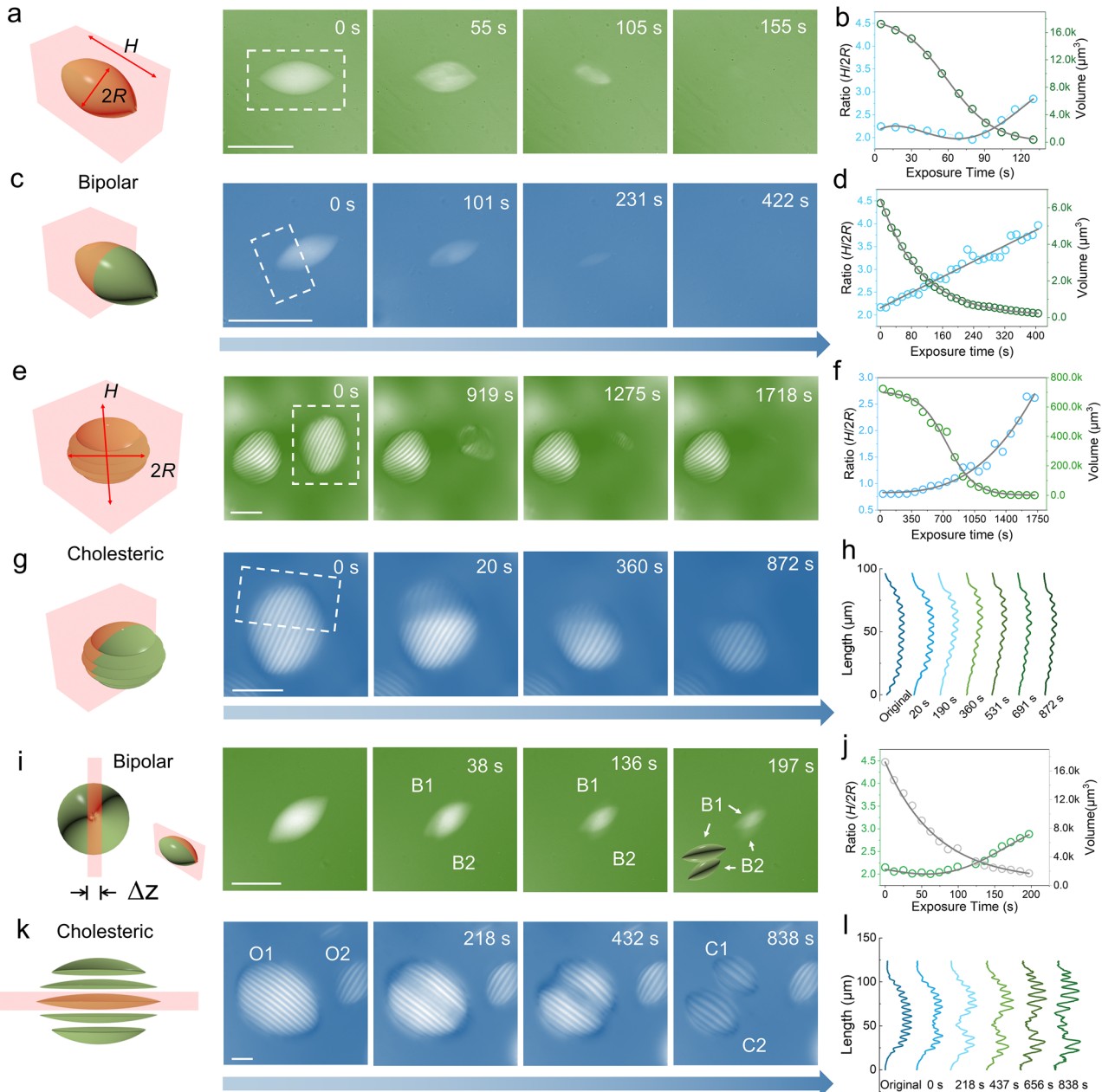

**Fig. 2 | Manipulation of amyloid tactoids by 3D-Heating. a.** Time-lapse images of irradiation of a single bipolar tactoid with a marked exposed region (fully covered). The scale bar is 50 μm. **b.** Aspect ratio and volume of fully exposed bipolar tactoids in (**a**) plotted vs. exposure time. The gray curves represent the fittings. **c.** Irradiation of half a bipolar tactoid. The scale bar is 50 μm. **d.** Aspect ratio and volume of half exposed bipolar tactoids in (**c**) vs. exposure time. **e.** Time-lapse images of a cholesteric tactoid being fully exposed to irradiation. The scale bar is 60 μm. **f.** Aspect ratio and volume of the cholesteric tactoid in (**e**). **g.** Exposure of half cholesteric tactoid. The scale bar is 60 μm. **h.** Dynamics of half pitch measured in the non-exposed region of the tactoid in (**g**). The color bar represents the value of half pitch. **i.** Break-up of a bipolar tactoid by 3D-Heating. The images show the cutting process of a large bipolar tactoid into two smaller bipolar tactoids, labeled B1 (inside) and B2 (outside). The thickness of the exposure region Δz = 5 μm. The scale bar is 30 μm. **j.** The aspect ratio and volume of the tactoid (B1) during the bipolar cutting. The gray curve represents the fittings. **k.** A cholesteric tactoid (O1) was cut into two smaller cholesteric tactoids along the band (2 R) with Δz = 15 μm. The resulting tactoids are labeled C1 and C2. The tactoid O2 is marked as the control experiment. The scale bar is 30 μm. **l.** Measurement of half pitch while O1 transforms to C1 and C2 tactoid. All of the images were obtained under a crossed polarizer. Models in panel **a, c, e, g, i** and **k** were created with 3ds Max 2021.

increased gradually from 6.6 ± 0.2 μm to 7.7 ± 0.2 μm, consistent with our general observation that smaller tactoids show larger half pitch values (Fig. 2h and Supplementary Table 1)[8]. Furthermore, the aspect ratio of the new tactoid increased from ~0.8 to ~1.2, evolving from an oblate to a prolate ellipsoidal morphology (Supplementary Fig. 7c)[16].

We further explored the potential of our 3D-Heating technique for manipulating tactoids. As shown in Fig. 2i, a "laser knife" was created to cut the bipolar tactoid into two separated bipolar tactoids. The laser

was scanned along the long axis (*H*) of the bipolar tactoid, with the length > *H*, width > 2 *R*, and the thickness of the laser box was set to 5 μm to ensure complete separation. This operation produced two stable bipolar tactoids with a 16 μm separation gap in solution at 197 s (Supplementary Fig. 7d). When tracking the dynamics of the internal structure of the new bipolar tactoid (B1 in Fig. 2i), an increase in the aspect ratio with respect to exposure time was observed (Fig. 2j). Roughly, we could draw the conclusion that the manipulation of

bipolar tactoids triggers the growing of aspect ratio, eventually even leading to the formation of homogeneous tactoids (Supplementary Fig. 3). Similarly, the cholesteric tactoid (O1) was cut into two smaller cholesteric tactoids using the laser knife along its band ($2R$) axis (Fig. 2k). After the cutting, the half-pitch value increased from $6.6 \pm 0.2$ µm to $9.8 \pm 0.1$ µm, providing direct evidence again that the half pitch increased in small cholesteric droplets (Fig. 2l and Supplementary Table 2). Notably, the cholesteric pitch values of neighboring tactoids, measured as the control experiment (see O2 in Fig. 2k) remained unchanged, confirming the resolution of our method in manipulating the tactoids. To further test the versatility of the "laser knife", we performed cuts at different operating directions where there was a 45° angle between the long axis ($H$) and the laser knife, as shown in Supplementary Fig. 8. The post-dynamic rearrangement of the separated bipolar tactoids was also recorded (Supplementary Movie 1). The beveled half tactoid reformed into a standard spindle-shaped bipolar within 400 s, providing insight into the phase recovery timescale of amyloid tactoids. In the case of cholesteric tactoids, cutting along the short axis ($H$) also yielded two separated tactoids (Supplementary Fig. 9), both of which showed increased half-pitch values. These results demonstrate the flexibility and precision of the 3D-Heating technique, offering a powerful tool for manipulating amyloid tactoids into a wide range of morphologies. Conversely, tactoids formed by cellulose nanocrystals (CNCs)[14,51], and exposed to the same laser dosage and conditions maintained their cholesteric structure without noticeable changes (Supplementary Fig. 10), as expected by the corresponding UV-Vis spectrum, which shows nearly no absorption at about 281 nm. This provides direct evidence that our 3D-Heating is highly specific to MPA materials, such as β-sheet-rich amyloid fibrils.

## Self-healing of amyloid tactoids

To investigate the LC phase recovery properties, a localized isotropic region was induced within a single tactoid using the 3D-Heating method. Owing to its high-resolution, as shown in Fig. 3a, a thin disk exposure region (red color) with a depth of 5 µm and a radius of 30 µm was created in the center of a cholesteric tactoid. Here, we found that the exposure area appeared as a shadow under a cross polarizer, indicating that a local isotropic phase was induced, but recovered quickly (Fig. 3b and Supplementary Movie 2). The shrinking speed of the disordered region revealed that the recovery along the POM bands ($S_l$, the length of the disordered region parallel to the cholesteric bands) was initially faster than the recovery perpendicular to the bands ($L_l$), as shown in Fig. 3c, indicating that mesogens realigned more easily along the bands where the mesogens were parallel to each other. Unexpectedly, from 20 to 40 s, 50 to 90 s, 100 to 130 s, and beyond 140 s, $L_l$ kept the same value and showed steps. These pinning effects strongly suggested that the process involved several transient energy minima, i.e., metastable states. It can be interpreted that the discrete behavior of $L_l$ relaxation is consistent with the number of bands (one band represents half a period of rotation of the mesogens). In contrast, the short axis ($S_l$) shrank rapidly within 40 s to a constant value, showing the largest ratio ($L_l/S_l$) of the isotropic region at 1.9 in Fig. 3d. Eventually, at around 183 s, the region transformed into a negative tactoid, maintaining an isotropic internal phase within the anisotropic medium, with the aspect ratio different than 1 and the $H$ aligned with the cholesteric bands. Furthermore, the recovery rate of the disordered area followed an exponential decay (Supplementary Fig. 11).

A further LC phase recovery study was carried out by the exposure of narrow rectangular areas with different orientations, as shown in Fig. 3e. Three narrow rectangular regions confining the isotropic phase were induced by 3D-Heating within a single cholesteric tactoid and designed with angles of 0°, 60°, and 90° relative to the cholesteric bands. Their morphological evolution (laser off) showed that isotropic regions with 0° maintained their orientation during recovery, while the

ones with 60° and 90° demonstrated rotational behavior and aligned with the same directions as the cholesteric bands after 409 s (Fig. 3f). A detailed analysis of the changes in the relative angles of the isotropic region (60°), similar to the results in Fig. 3c, revealed that the angle changes occur in steps at ~40 s, ~65 s, and 90 s (Fig. 3g). In other words, the process of healing passed through metastable states associated with cholesteric bands during the path towards the full recovery of the ground-state LC configuration. This stepwise behavior was confirmed in the analysis of the time evolution of the length of the narrow rectangular regions in Fig. 3h. In addition, the recovery speeds for isotropic regions demonstrate a clear behavior that the region parallel (0°) to the bands shows the fastest shrinking speed, while the orthogonal cut (90°) has the slowest speed. Finally, the length of three isotropic regions stabilized at ~20 µm after 120 s. Further study with a cross pattern and spots array pattern in Supplementary Fig. 12 additionally confirmed that the recovery of the isotropic region parallel to the bands was quicker than in the vertical direction, resulting in a negative tactoid with the long axis parallel to the bands.

We further monitored the evolution of the laser-induced negative tactoid. Supplementary Fig. 13 shows a negative tactoid crossing three bands in the center of a cholesteric tactoid, induced by the 3D-Heating method. The negative tactoid faded after 308 s, with full vanishing (LC reconstruction) completed in 1,000 s, a process nonetheless accompanied by a slight reduction in the total volume of the cholesteric tactoid. Additionally, as the negative tactoid shrank, the recovery rate increased. The half pitch, which broadened during laser irradiation, exhibited a shrinking behavior when rebuilding the cholesteric structures in the absence of the pulsed laser. Interestingly, from the monitoring of laser-induced negative tactoid arrays (Supplementary Fig. 14), the isotropic regions tended to fuse with their neighboring regions along the $2R$ direction (the fibrils were parallel to each other), but never combined in the $H$ direction (the fibrils were rotated to each other). When the laser-induced isotropic region was large (50 µm in diameter) and its depth reached ~10 µm (near the pitch value), we captured the entire localized nucleation process (Fig. 3i). The cylindrical isotropic region evolved into a negative spindle-like shape after 36 s. A cholesteric nucleus formed at 142 s in the center of the isotropic region (Supplementary Movie 3). Although the cholesteric nucleus had an initially different director compared to the surrounding cholesteric boundary, it merged and connected with the surrounding structure in subsequent steps, completing the rebuilding process in 1,052 s. These results directly confirmed that amyloid fibril tactoids possess a strong self-healing ability to restore the LC ground state via a localized process of nucleation and growth.

In order to understand the self-healing ability of the tactoids under conditions of extreme, disruptive deformation, experiments were designed in which half of the cholesteric tactoids were ablated using a higher laser power (~0.3 nJ per pulse). Two cholesteric tactoids with different orientations ($H_1$ and $H_2$) were selected according to the model illustrated in Fig. 3j. Half of both tactoids were exposed to the laser at the same laser exposure. POM images exhibited two half-shaped cholesteric tactoids in the beginning (marked as 0 s in Fig. 3j) after laser irradiation. Importantly, we captured the rapid formation of cholesteric structures on the cutting face (cross section) within the next 222 s. The two half tactoids recovered into new prolate tactoids, with an aspect ratio ($H/2R$) larger than 1 (Supplementary Fig. 15 and Supplementary Table 3), which demonstrates the ultrafast and accurate manipulation by the 3D-Heating method and remarkable self-healing features of the tactoids even under extreme deformation conditions.

## Operation in nanoparticle anchored tactoids

The 3D-Heating method provides us with the opportunity to build complex colloidal assembly structures and patterns. As a proof of concept, amyloid tactoids were functionalized with fluorescent

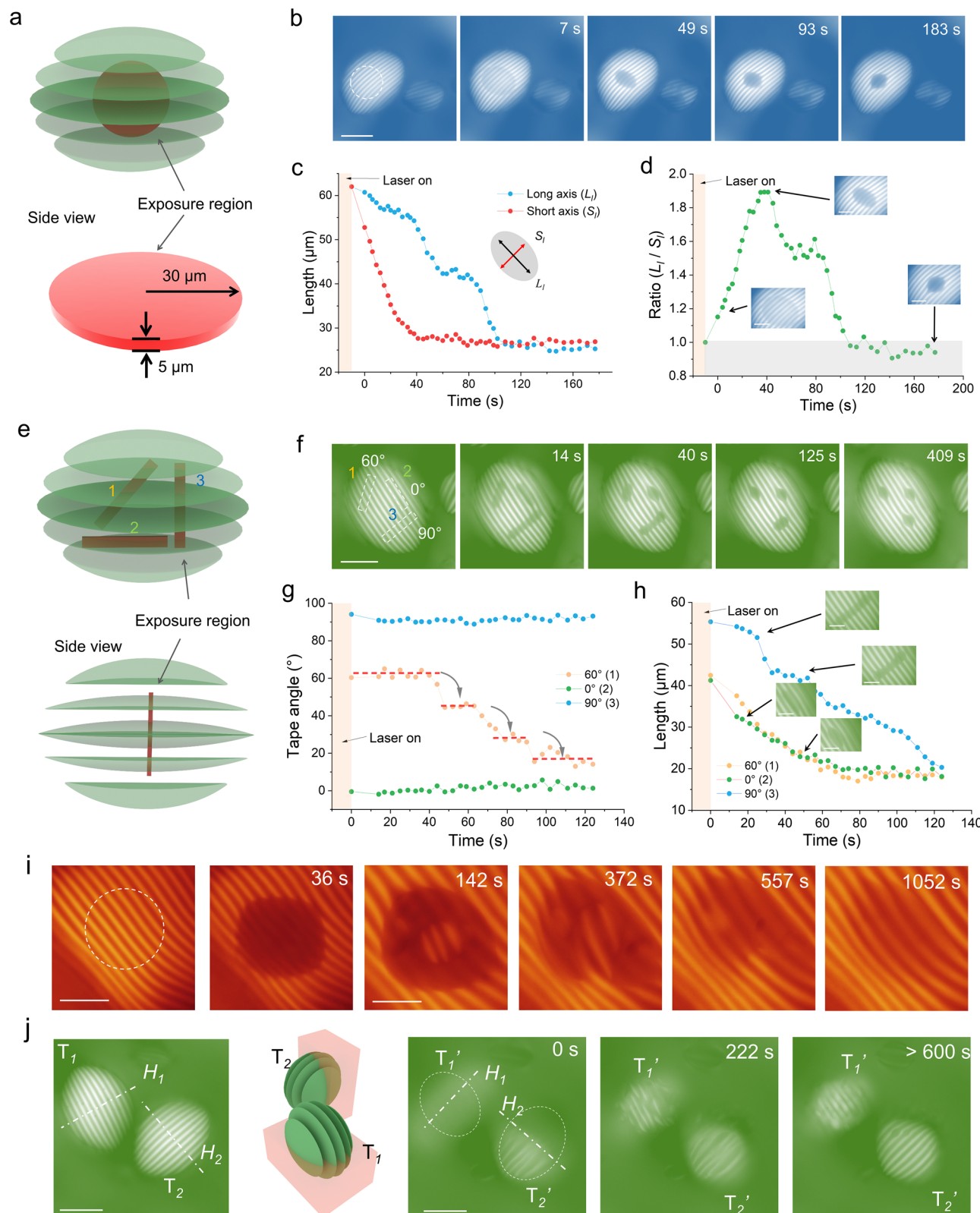

nanoparticles ($d$ = 30 nm) to create fluorescent-LC hybrid droplets. As shown in Fig. 4a, the nanoparticles became self-enriched in the cholesteric tactoid, whereby LC properties were fully preserved (as evidenced by the bands) under confocal fluorescence microscopy. In contrast to previous studies on fluorescence-labeled tactoids, in which fluorescent signals were uniformly distributed within the cholesteric tactoid[6,9], the bright and dark band phenomenon suggests that the

nanoparticles aligned with the amyloid mesogens in an orderly manner. Additionally, the fluorescent nanoparticles enabled 3D-Imaging of the hybrid droplets (Supplementary Fig. 16). For example, we could quickly monitor the tactoid along the $xy$ and $xz$ axes (which cannot be measured in POM) in situ and perform 3D modeling of the tactoid, enabling precise monitoring of the internal structures of tactoids (Fig. 4b and Supplementary Movie 4).

**Fig. 3 | The recovery of cholesteric tactoids after locally printed patterns. a.** Schematic of an exposure region (disk shape with Ø=60 μm and $h$ =10 μm, red color) in the center of a tactoid. **b.** The changes in a laser-induced isotropic region over time within a cholesteric tactoid. The scale bar is 60 μm. **c.** The changes in the length of the long axis ($L_l$, blue) and the short axis ($S_l$, red) of the isotropic region are shown in panel (**b**). **d.** The dynamics of the aspect ratio ($L_l/S_l$) of the isotropic droplet. **e.** Schematic of the exposure region with three rectangular regions, marked with 1, 2 and 3. **f.** The recovery dynamics of three laser-induced isotropic regions having 60° (1), 0° (2), and 90° (3) to the long axis (2 $R$), respectively. The exposed regions maintain the same area initially. The scale bar is 50 μm. **g.** Angle changes in the three narrow tape-shaped regions from (**f**). **h.** Length of three narrow tape-shaped regions vs. recovery time. **i.** Locally induced isotropic zone (negative tactoid) in the central region and its recovery process, highlighting the nucleation dynamics. The tactoid fully recovered after ~1052 s. The scale bar is 20 μm. **j.** Schematic illustration showing the ablation of half a cholesteric tactoid in two directions, resulting in two independent tactoids (T1 and T2). Time-lapses show the recovery of the cutting boundaries of $T_1$ and $T_2$. The scale bar is 50 μm. All of the images were captured under a crossed polarizer. Models in panel **a**, **e**, and **j** were created with 3ds Max 2021.

Combining the 3D-Heating and 3D-Imaging methods, as shown in Supplementary Fig. 17, six ordered isotropic points were printed inside the hybrid tactoid spatially, which gradually recovered to the cholesteric phase one by one after 956 s. Here, with the help of $xz$ axes scanning, we show that six isotropic points recovered to the cholesteric phase from ~284 s to ~956 s (Supplementary Fig. 17b). We further show that patterns could be printed on a single layer (exhibiting a homogeneous fluorescence signal) in 3D space. As a demonstration, the *ETH* logo was printed on one of the bright bands in the hybrid tactoid (Fig. 4c). The pattern evolved first into a negative tactoid within 356 s and then fully recovered to the ground LC state after 1,051 s. Upon examining the fluorescence intensity in the bright layer (Fig. 4d), the logo region revealed an increasing average intensity as it returned to the cholesteric phase. However, the intensity in the surrounding area (above and below the logo) decreased slightly in approximately 400 s, but recovered after 1051 s. At the same time, the area of the operated layer recovered with a slight decrease in the area (Fig. 4e), thus providing full evidence that the large-scale reconstruction of the cholesteric structure is governed by fast dynamic and cooperative nature.

To showcase the strong manipulation capabilities of 3D-Heating and the tactoid's self-healing features, the tactoid could be separated into three (Fig. 4f and Supplementary Movie 5) or multiple pieces (Supplementary Fig. 18 and Supplementary Movie 6), and then quickly recovered within 353 s and 411 s. The half pitches were dynamically monitored, shrinking quickly in the first 75 s and expanding in a stepwise manner during the subsequent process (Fig. 4g). Notably, two key steps occurred when the separated tactoids began reconnecting. Additionally, the length of 2 $R$ increased to 93 μm and then returned to its original value after 340 s (Supplementary Fig. 19). To better understand the dynamics of connection, one cholesteric tactoids was selected and cut along both horizontal and vertical directions consecutively (Fig. 4h). First, cutting along the bands (2 $R$) created an open zipper, which closed rapidly and completed in 116 s (see Supplementary Movie 7). The zoomed image of separated tactoids is shown in Supplementary Fig. 20a. Similar to the decrease of area in the logo printing (Fig. 4e), the cutting of the tactoids along the bands also shows the reduction of 2 $R$ (Supplementary Fig. 20b). In contrast, the $H$ has a temporary increase under equilibrium. When the cutting direction was rotated 90° and operated on the same tactoid that recovered to its original shape, the zipper took much longer to close - starting at 165 s and finishing at 387 s (Fig. 4i and Supplementary Movie 8). This finding strongly suggests that the recovery of the original cholesteric structure is more difficult when mesogen rotation is involved across several bands. Unlike the cutting along the bands, 2 $R$ demonstrated a temporary increase under recovery, while the $H$ decreased accordingly (Supplementary Fig. 21). As a result, the cutting along the different directions results in markedly different recovery dynamics. Additional laser manipulations were also performed. From the random cutting of tactoids in Supplementary Fig. 22, the tactoids can be constructed into flexible patterns. However, all of the patterns exhibited strong self-healing abilities and recovered to their original shapes. The sub-micron resolution of our laser operation is best showcased in Supplementary Fig. 23, where a 1.4 μm length gap was created to separate the cholesteric tactoid into two parts.

## Pitch-tunable amyloid tactoids

Our 3D-Heating method can be applied not only for the cutting and self-healing of individual tactoids, but also for controlling their internal structure. For example, tunable pitch control is crucial for optical applications, such as producing structural color and circularly polarized luminescence[52,53]. When the laser was set to a moderate power (~0.2 nJ per pulse), we found that the irradiation could first trigger the melting of the local order, and then induce a new band in the exposed region. As illustrated in Fig. 5a, the half pitch length expanded under exposure, and new periods were induced, i.e., a new light band appeared in each laser-exposed region. Our experimental results in Fig. 5b demonstrated that a 10-band cholesteric tactoid was rebuilt into a 12-band tactoid by triggering the onset of the new bands. The length ($H$) of the tactoid increased dramatically from the original length of 65.7 μm to 98 μm. As illustrated by the fluorescence intensity evolution map (Fig. 5c and Supplementary Fig. 24a), the exposed tactoid exhibited a relaxation period of 150 s, during which the tactoid grew from 94.3 μm to 98 μm. As the tactoid stabilized (shrank) into a steady state, the formation of new bands was observed in the relaxation regions between 110 and 150 s (Fig. 5b). At this point, as marked with white arrows in Fig. 5c, new bands appeared and equilibrated with the surrounding cholesteric structures. The final length ($H$) of the tactoid shrank to 76 μm after 500 s, and the half-pitch sizes remained heterogeneous; the newly formed bands revealed half-pitch values between 5.6 and 6.1 μm (Supplementary Fig. 24b), smaller than the original one (~7.5 μm) (see Supplementary Movie 9). Notably, not only did the number of bands change, but the half-pitch length was also reduced.

The 3D-Heating method may also allow for erasing the bands. As shown in Fig. 5d, the newly built 12-band tactoid was engineered into tactoids with 11, 10, and 9 bands through step-by-step laser irradiation. In this case, a relatively high laser power (~0.3 nJ per pulse) was applied to dissolve the mesogens into the surrounding isotropic environment, achieving the new stable structure with the desired number of bands after a 10-min equilibrium. Moreover, this method can also fix structural dislocation in tactoids, as shown in Supplementary Fig. 24c. A two-phase (heterogeneous) tactoid was restored to a single-phase tactoid after laser exposure in the marked region. To further explore this function, we exposed the tactoid to low-power irradiation (~0.14 nJ per pulse) in dark bands. The bands expanded without the formation of new bands and returned to their original state after ~300 s, as illustrated by the fluorescence intensity map in Fig. 5e over five cycles. The half pitch was tuned between ~5.6 μm and ~7.5 μm (Fig. 5f). In aggregate, and similarly to a micron-blade scalpel, we were able to precisely manipulate the local half-pitch of the chiral nematic bands, which may be of paramount importance for optical applications exploiting cholesteric phases.

Finally, due to the strong self-healing features and the high degree of control over the 3D-Heating process, the fluorescent amyloid hybrid tactoid was utilized as a smart optical switch, equipped with self-healing and light converting functionalities (Fig. 5g). The rotation of the tactoid (0-360°) exhibited high optical response, switching between ON and OFF states based on light intensity (Fig. 5h). As an advantage compared with the POM image, the fluorescence imaging

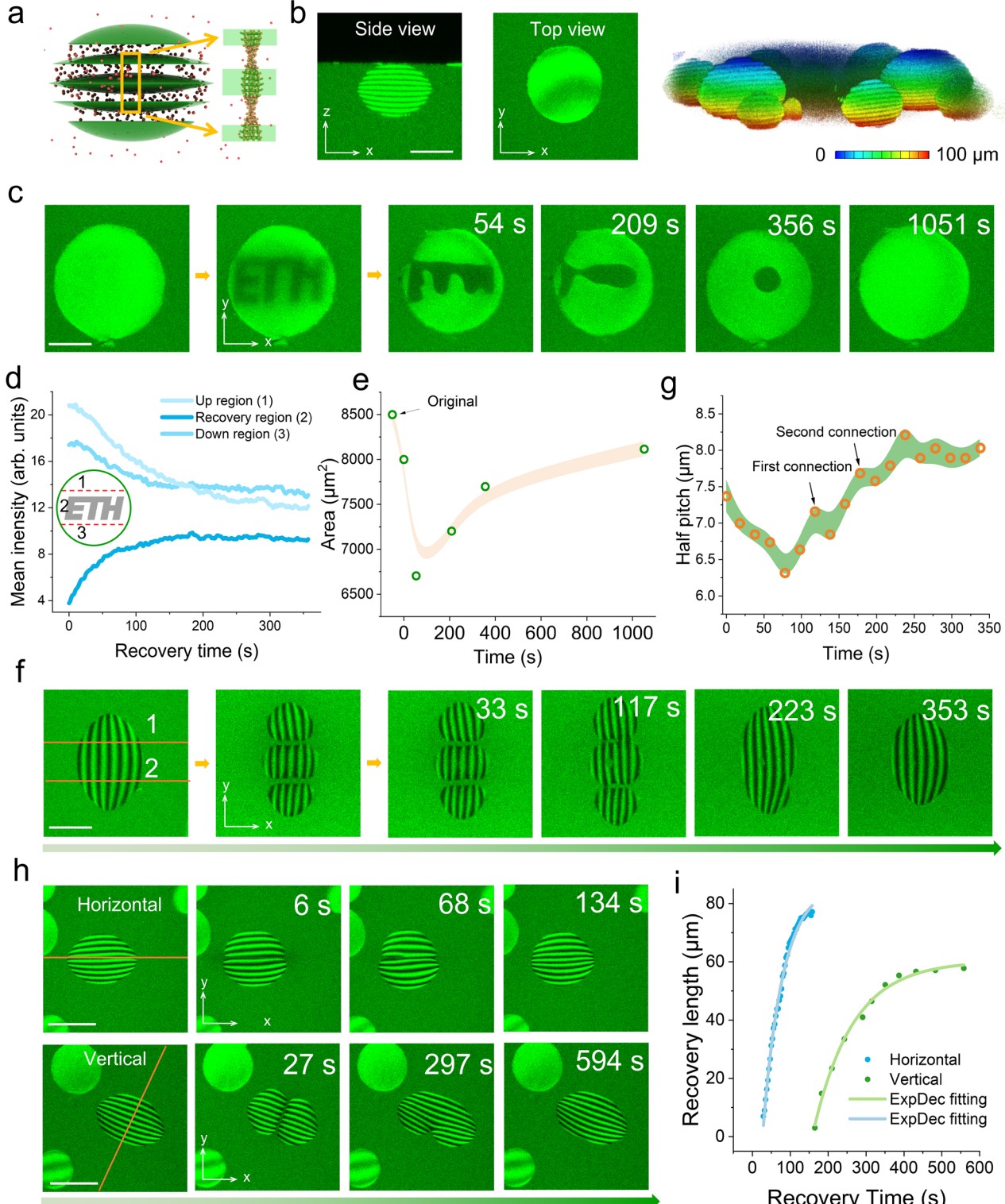

**Fig. 4 | Templating of fluorescent nanoparticles by amyloid cholesteric tactoids and the strong self-healing of hybrid tactoids. a.** Illustration depicting the distribution of PS nanobeads (30 nm in diameter) templated by cholesteric tactoid. The nanoparticles are aligned along the chiral amyloid fibrils, forming fluorescent bands observable under fluorescence confocal microscopy. **b.** The *xy-* and *xz-* axis of fluorescence images of single nanoparticles-doped cholesteric tactoid, with their 3D structure by the *z*-stacking method. The scale bar is 60 μm. **c.** Direct printing of a pattern (*ETH* logo) on a cholesteric hybrid tactoid and its dynamic self-healing process. The scale bar is 40 μm. **d.** The records of fluorescence intensity in the logo area and surroundings (up and down region) of the hybrid tactoid in **c. e.** The

dynamics of the area from the printed droplet. The data were expressed as mean ± standard deviation (S.D). **f.** The hybrid tactoid was cut into three pieces, followed by a rapid self-healing process to a normal cholesteric tactoid. The scale bar is 30 μm. **g.** Dynamics of half pitch from the middle fragment in **f** during the self-healing process. The data were expressed as mean ± standard deviation (S.D). **h.** Two types of cutting (along the bands and perpendicular to the bands) with respective rapid recovery processes were recorded. The scale bar is 60 μm. **i.** The self-healing length of the cutting hybrid tactoid vs. recovery time in **h.** Models in panel **a** were created with 3ds Max 2021.

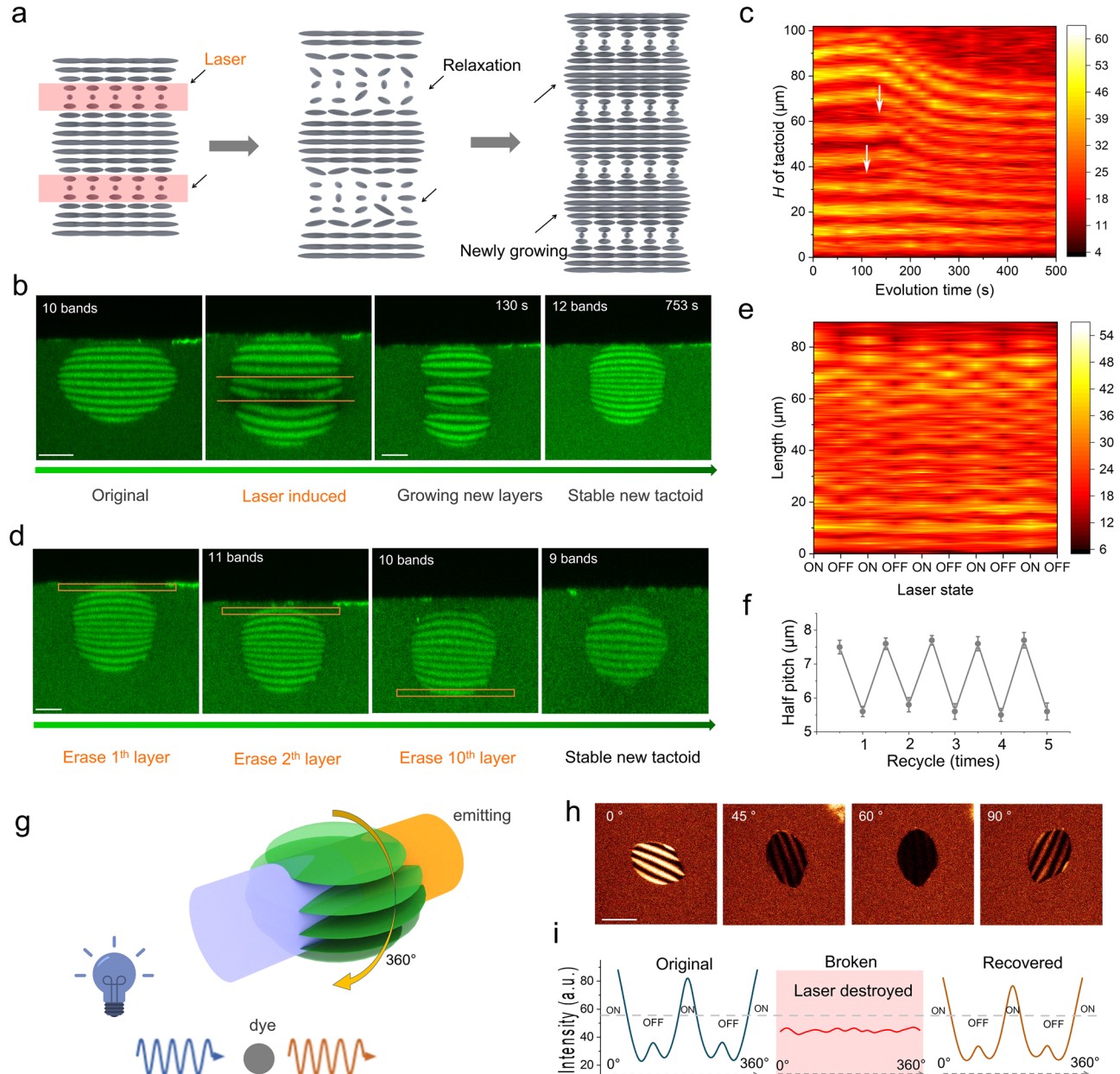

**Fig. 5 | Control of half pitch in hybrid tactoids using 3D-Heating. a**. Description of cholesteric dark bands exposed to laser irradiation, and their relaxation and growth of new bands. **b**. Exposure of two dark bands (No. 4 and 6) and their evolution. Two new bands emerged after 753 s, as captured by side view fluorescence microscopy. Scale bars, 25 μm. **c**. Fluorescence intensity map recorded from **b**. Two white arrows represent the initiation of newborn bands. **d**. Erasing of bands on the tactoid from **b**. The marked region was exposed with the laser. We erased the 1st, 2nd, and 10th bands sequentially. Scale bars, 25 μm. **e**. Fluorescence intensity map of length (*H*) during cycles of laser exposure (laser power of ~0.14 nJ per pulse), with alternating laser ON and OFF. Each cycle lasted 5 min. **f**. Average half-pitch values recorded during the laser exposure cycles are shown in **e**. The data were expressed as mean ± standard deviation (S.D). **g**. Schematic diagram showing the rotation (360°) of a tactoid under the laser. The emitting light was tunable with the dye properties. **h**. The fluorescence images of different rotation angles were obtained using confocal fluorescence microscopy. **i**. The optical intensity was captured with the rotation of tactoid from 0° to 360°. Then, the focused region was erased by 3D-Heating. The repeated recording from 0° to 360° was finished after 5 min to wait for the self-healing of the tactoid. Scale bar, 30 μm. Models in panels **a** and **g** were created with 3ds Max 2021.

exhibits a stable background intensity (Supplementary Fig. 25). In this case, the rotation of the tactoid produced a fluorescence signal that clearly followed an ON-OFF-ON-OFF-ON sequence, while the emitting light color was dependent on the dyes. When the focused layer of the tactoid was selectively erased by the laser (Supplementary Fig. 26), the switching capability vanished, resulting in a flat-line response (Fig. 5i). After 10 min, however, the switch regained functionality as the cholesteric structures recovered. This simple example demonstrates how fluorescent signals can also be precisely engineered by combining LLCPS, fluorescent probes, and the 3D-Heating process.

## Discussion

We have leveraged the nonlinear absorption features of amyloid fibrils under pulsed laser and the anisotropic liquid crystalline properties of amyloid tactoids to demonstrate that MPA can be applied to locally and spatially manipulate tactoids at the single-droplet level, and with a sub-micrometer 3D resolution, introducing a tunable external stimulus to direct self-assembly in heterogeneous liquid crystalline systems. We have shown that a single amyloid tactoid can be spatially manipulated into different morphologies, and that locally melted isotropic phases within a single cholesteric tactoid can recover their ground LC state

within minutes, exhibiting a strong memory effect and self-healing properties. By controlling the laser power density, cholesteric tactoids can be restructured into new tactoids with a different number of cholesteric bands and periodicities, or cholesteric bands can be erased and ablated one by one as desired. Exploiting these functions, we presented a simple model of an optical switch enabled by our 3D-Heating method.

On a purely fundamental level, the possibility to locally melt the nematic state into the isotropic state under the control of a laser and recover it reversibly indicates that LLCPS in the present case cannot be a purely entropic process, as predicted by the Onsager theory and widely believed. The reversible, laser-induced transition between nematic and isotropic phases indicates a temperature-dependent shift in the free energy landscape. This suggests that enthalpic contributions are at play, beyond the purely entropic mechanisms predicted by Onsager's theory. More generally, this work also demonstrates that it is possible to use non-invasive local stimuli to control phase separation-within-phase separation, opening original possibilities in the thermodynamic hierarchical control of complex fluids.

On a more applied level, these results may open the floor to potential applications in engineering, printing, memory-storage, and self-healing in nanotechnology, life sciences, and biomaterials design. For example, the results presented here clearly show how all of the salient features of localized LLCPS can be tuned, controlled, and engineered at a scale of individual droplets (and within), with a submicron 3D resolution, departing markedly from the existing methods to control the morphology in phase-separated heterogeneous colloidal systems. Additionally, the incorporation of fluorescent probe particles facilitates the printing and machining of information, which can be read simultaneously or individually using both cross-polarized and fluorescent light, ultimately establishing unexplored tools for innovative optical sensors and devices.

## Methods

### Preparation of amyloid tactoids

The BLG amyloid fibrils were prepared using the β-lactoglobulin protein (whey protein). β-lactoglobulin protein powder was dissolved in Milli-Q water, and the pH of the suspension was adjusted to 2. Then, the suspension (2 wt.%) was filtered with a nylon syringe filter (0.45 μm, Huberlab) to remove protein aggregates. In order to obtain the amyloid fibrils, the solution was heated at 90 °C for 8 h. Amyloid fibrils were then cut into short fibrils by mechanical shear forces. Next, a dialysis was performed for 7 d using a 100 kDa (MWCO) Spectra/Por dialysis membrane (Biotech CE Tubing) with daily bath replacement. The final concentration of short BLG fibrils with an average length of ~435 nm was achieved by up-concentrating the solution with the reverse osmosis method, in which a dialysis membrane (standard RC tubing) was used against a 10 wt.% polyethylene glycol (Sigma Aldrich) solution under pH = 2. Phase separation of the amyloid fibril suspension was found after 3–4 weeks when stored in bottles.

All glass capillaries and containers were cleaned by piranha solution, ethanol, and Milli-Q water. The carboxylate-modified latex bead solutions ($\lambda_{ex}$ ~ 520 nm; $\lambda_{em}$ ~ 540 nm, Sigma Aldrich) were centrifuged twice at 16500 ×g and replaced with pH = 2 Milli-Q water. The BLG fibril suspension was used at a concentration in the coexistence region of the isotropic-nematic phases and incubated in a capillary tube with the size of 50 mm × 4 mm × 200 μm. For fluorescent tactoids, a designed volume (2 ~ 5 μl, 500 nm, 1 %) of latex bead solution mixed with the BLG suspension (34 - 37 μl) was injected into the capillary tube gently by pipette. Finally, both ends of the capillary tube were sealed with UV-curable glue. The equilibrium was performed at room temperature.

### Multiphoton heating

Experiments were performed using a multiphoton microscope with Mai Tai XF (Spectra-Physics) femtosecond laser, tunable from 720–950 nm. Ultrashort (100 fs) laser pulses at a pulse repetition frequency of 80 MHz were focused into the bulk of the amyloid tactoids using a 25× 0.95NA L Water HCX IRAPO microscope objective. The scheduled process is shown in Fig. 1. A capillary tube with well grown tactoids was put on the scanning platform of an inverted confocal fluorescence microscope (Leica SP8). The microscope equipped with a polarizing filter was used to check the tactoids without fluorescence dyes. The switch between femtosecond laser exposure and fluorescence imaging had a 7–10 s delay in the full frame scanning.

The microscope can sequentially scan a single point or a small region of interest (ROI) within a sample. The full frame resolution (laser points) was set as 1024×1024 or 512×512. As a result, ROI scanning provides us with the possibility to scan designed patterns. Both the laser beam can be moved orthogonally to the $xy$ plane by high-speed oscillating mirrors and the sample along the z axis by a z piezo stage. The single scan speed is 5–7 full frames per second.

### Polarized optical microscopy and fluorescence microscopy

Cross-polarized optical microscopies (Zeiss Axio Imager M1m and Leica SP8) were used to record the images of the tactoids. The microscope (Zeiss Axio Imager M1m) equipped with LC (liquid crystal)-PolScope was used to capture the colorful PolScope images of the tactoids. To obtain the multiphoton image and three-dimensional fluorescent image, the multiphoton excitation confocal microscopy (Leica TCS SP8) with a water-immersed 25× objective was used. The excitation source used in our experiments was a continuous laser at 561 nm and a femtosecond laser at 780–950 nm. The confocal microscopy was also used to record the polarized image by an additional polarized filter. All analyses were performed using ImageJ, Zen software, and Laica X software.

### Atomic force microscopy

The amyloid fibril samples were diluted to a final concentration of 0.02 wt.% to prepare for scanning by AFM. An aliquot (5 μL) of the diluted solution was deposited on a freshly cleaved mica, followed by rinsing with Milli-Q water and drying with a gentle compressed air flow after 3 min. A multimode VIII scanning probe microscope (Bruker) with a commercial silicon nitride cantilever (Bruker) at a vibration frequency of 150 kHz was used to obtain the morphologies. Finally, AFM images were simply flattened using the Nanoscope software (Bruker). The histogram of the fibril length was fitted with the Gaussian distribution.

### Absorption spectrum and z-scan spectrum

The fabricated samples were diluted to a final concentration of 0.2 mg/mL in a 10 mm glass cuvette for the absorption spectrum measurements (JASCO spectrophotometer). Z-scan experiments were carried out using a laser system consisting of a femtosecond pulse generated by an amplified femtosecond system at an 80 MHz repetition rate, ~100 fs pulses operating as an 810 nm pump. Amyloid solutions were placed in a 1 mm deep glass capillary tube sealed with UV glue prior to scanning. The laser was focused onto the sample by a lens to a minimum beam waist radius of 30 μm measured by the knife edge method. The narrow path length cuvette ensured that the length was shorter than the Rayleigh length of the beam (≈3.5 mm). A silicon photo detector (Thor Labs, Newton, NJ, U.S.A.) was used to collect signals. Samples were driven along the beam path at a constant speed of 2 mm/s by an electronic control platform. Z-scans were collected from three independent positions in a capillary tube.

## Data availability

All data that support the findings of this study are available within the article and the Supplementary Information. Source data are provided with this paper.

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

## Acknowledgements
This work was supported by ETH Zurich, the Fundamental Research Funds for the Provincial Universities of Zhejiang, the National Natural Science Foundation of Ningbo (No. 2024J198), the National Natural Science Foundation of Zhejiang (No. MS26A040024), the Qian Xuesen Collaborative Research Center of Astrochemistry and Space Life Sciences Fund, and the K. C. Wong Magna Fund in Ningbo University. The authors thank Dr. Justine Kusch and Prof. Weidong Tao, who provided great assistance in multi-photon technology. The authors also express gratitude to members of our laboratories for helpful discussions, and Dr. Ye Yuan for invaluable comments on the manuscript. Y.Z. acknowledges the support from the National Natural Science Foundation of China (Nos. 42388101 and 92256203) and the Ningbo Top Talent Project (No. 215-432094250).

## Author contributions
D.L. and R.M. conceptualized and designed the study. D.L. performed the experiments. D.L., H.A., Y.Z., and R.M. analyzed the results. D.L., H.A., and R.M. wrote the manuscript. Y.Z. and R.M. supervised the project. All authors participated in the revision of manuscript.

## Funding

## Competing interests
The authors declare no competing interests.
