## [Transparent Peer Review file · Nature Communications]

Remodeling and Self-Healing of Individual Amyloid Tactoids via Multiphoton Absorption

Corresponding Author: Professor Raffaele Mezzenga

Version 0:

Reviewer comments:

Reviewer #1

(Remarks to the Author)

This manuscript reports laser-induced manipulation of amyloid tactoids formed by isotropic-nematic transition in colloidal systems of anisotropic particles. Various types of manipulations such as local melting, cutting, and self-healing are examined. Although the results are predictable and very reasonable, direct observations and experimental analysis with optical microscopy are new and valuable. For these reasons, I recommend publication after some revisions suggested below.

- Various types of manipulations shown in the manuscript should require different conditions of laser-beam irradiation. Laser power should be the most important but other factors such as irradiation area can also contribute. Kinetic reasons such as appropriate irradiation period are also possible. Thus, summary of the relationship between the manipulation type and laser conditions is very valuable for readers.

- Recover of the nematic structure after the perturbation by the laser irradiation is reasonable because thermodynamic stability of the nematic state. However, the tactoid reformation would not be explained by thermodynamics. Non-recoverable damages might happen under some irradiation conditions.

- In the Discussion section, the authors concluded the contribution of enthalpic process in the tactoid formation and manipulation against the Onsager theorem assuming entropic process. For my understanding, however, the Onsager theorem does not predict the tactoid formation but only explains the isotropic-nematic transition. Also, many experimental studies have demonstrated that real systems do not rigidly obey the theorem. Contribution of non-entropic, i.e. enthalpic process should not be a new finding. In that sense, the results should be discussed in relation to the formation mechanism of tactoids.

- Molecular mechanism of the tactoid collapse, reformation, and structural modification should be given.

Reviewer #2

(Remarks to the Author)

This work describes manipulation of amyloid colloidal liquid crystalline structures, tactoids, by micron scale targeted heating with a laser. The efforts done to visualize these phenomena as images, in short time scale, are very impressive and results highly interesting to learn from.

Effects due to manipulation have been demonstrated from multiple angles making it interesting from application point of view. The different responses between bipolar and cholesteric tactoids to manipulation are very well documented and explained for further utilization.

It is curious to think that the 3PA technic would work on cellulose nanocrystal (CNC) tactoids as well, but since there's no aromatic functionalities, should the CNC material be somehow functionalized (chemically or with additives) to probe the

technic on CNCs?

I recommend publishing as it is. Beautiful work!

Reviewer #3

(Remarks to the Author)
review attached as pdf

Version 1:

Reviewer comments:

Reviewer #1

(Remarks to the Author)
I satisfied the revision of the manuscript. I recommend publication as is.

Reviewer #2

(Remarks to the Author)
Nice work, I recommend publishing.

Reviewer #3

(Remarks to the Author)
The authors have adequately addressed all the reviewer's comments and concerns. The reviewer notes that the other reviewers' assessments were similarly positive, which reinforces that this manuscript is suitable for Nature Communications. The manuscript is now ready for publication.

Responses to Reviewers

NCOMMS-25-63118-T (Writing, reading, erasing, cutting and self-healing individual amyloid tactoids by multiphoton absorption)

Reviewer #1

This manuscript reports laser-induced manipulation of amyloid tactoids formed by isotropic-nematic transition in colloidal systems of anisotropic particles. Various types of manipulations such as local melting, cutting, and self-healing are examined. Although the results are predictable and very reasonable, direct observations and experimental analysis with optical microscopy are new and valuable. For these reasons, I recommend publication after some revisions suggested below.

Response: We thank the Reviewer for their positive comments on our work. We addressed all the suggested revisions in the revised manuscript.

Comment 1: Various types of manipulations shown in the manuscript should require different conditions of laser-beam irradiation. Laser power should be the most important but other factors such as irradiation area can also contribute. Kinetic reasons such as appropriate irradiation period are also possible. Thus, summary of the relationship between the manipulation type and laser conditions is very valuable for readers.

Response: We agree with the Reviewer, and in the revised manuscript, we added the following section to the SI to clearly explain the relationship between the manipulation type and laser conditions.

In our experiments, we applied irradiation using Point Scanning Confocal Microscopy, which employs a focused laser beam to sequentially scan a single point or small region of interest (ROI) within a sample. The full frame resolution (laser points) was set to 1024×1024 or 512×512 , where the ROI feature allows for scanning designed patterns. To show the relation between the irradiated area and the actual area affected within the sample, we performed the following experiments. We designed three different irradiation areas (48.6, 176.4 and $360.6 \mu\text{m}^2$) in ROI as shown in

Supplementary Fig. 5 and irradiated them under the same conditions. Following the irradiation, we immediately measured the disordered regions within the sample, marked with red dash circles in Supplementary Fig. 5b. Our measurements show that the area set by the ROI and the disordered region measured within the samples are of comparable size, indicating the same irradiation effects on different areas. We note that the slightly lower values for disordered areas compared to the ROI region are due to a measurement delay (> 10 s) between the NIR irradiation and POM imaging, which is unavoidable for technical reasons.

Additionally, to show how the kinetics of the irradiation plays a role in disordering the ordered amyloid fibril samples, we performed two sets of experiments with different exposure time intervals, but the same total irradiation time (Supplementary Fig. 5d). In the first set of experiments, a sample was irradiated with a pixel dwell time of $1.2 \mu\text{s}$ to obtain an isotropic region ($167.1 \mu\text{m}^2$). In the second set, the pixel dwell time was changed to $0.4 \mu\text{s}$ with three repetitions, and the system was given the same amount of time to recover after each exposure. The result shows that, despite the same total exposure time (and thus energy), the isotropic region obtained by laser with recovery intervals was smaller ($121.9 \mu\text{m}^2$) than that from a single continuous exposure.

The discussion was added in revised manuscript and highlighted with yellow color as follows:

“Systematically, the relationship between the laser-induced nematic-isotropic transition rate and laser power was investigated. As shown in Fig.1f, we found that a power of 13.6 mW (total pixel dwell time of $16 \mu\text{s}$) could have distinct breaking effect on cholesteric structures. Here, to show the relation between the irradiation area and the actual area affected within the sample, we designed three different irradiation areas (48.6 , 176.4 and $360.6 \mu\text{m}^2$) as shown in Supplementary Fig. 5 and irradiated them under the same conditions. Following the irradiation, we immediately measured the disordered regions within the sample, marked with red dash circles in Supplementary Fig. 5b. Our measurements show that the area set by program and the disordered region measured within the samples are of comparable size (Supplementary Fig. 5c), indicating the same irradiation effects on different areas. We note that the slightly lower

values for disordered areas compared to the set region are due to a measurement delay (> 10 s) between the NIR irradiation and POM imaging. Additionally, the kinetic irradiation was also measured in Supplementary Fig. 5d. The result shows that the isotropic region ($121.9 \mu\text{m}^2$) obtained by ON-OFF laser period (pixel dwell time of $0.4 \mu\text{s}$ with three repetitions) was smaller than the single continuous exposure ($167.1 \mu\text{m}^2$, pixel dwell time of $1.2 \mu\text{s}$). Over all, this method provided us with a powerful and versatile method for generating complex 3D disordered phase patterns within the otherwise LC-ordered state of the tactoid.”

Supplementary Fig. 5 | The effects of irradiation area and time on samples of ordered amyloid fibrils. a. Three different irradiation areas were designed. **b.** The ensued disordered regions within the sample as a consequence of irradiation are marked with red dashed circles. Scale bar is $30 \mu\text{m}$. **c.** The irradiation area and the resulting disordered region (isotropic area) measured within the samples are of comparable size.

d. The comparison between a sample irradiated with a pixel dwell time of 1.2 μs and a sample irradiated with the same total exposure time, but divided into three times with recovery intervals. Scale bar is 10 μm .

Comment 2: Recover of the nematic structure after the perturbation by the laser irradiation is reasonable because thermodynamic stability of the nematic state. However, the tactoid reformation would not be explained by thermodynamics. Non-recoverable damages might happen under some irradiation conditions.

Response: We first note that, in our experiments, we set the laser irradiation power to be less than 20 mW, which does not result in non-recoverable damage in the amyloid fibrils (Fig.1). Previously, with a series of femtosecond laser irradiation experiments (*ACS Nano* 2023, 17, 10, 9429–9441; *ACS Chem. Neurosci.* 2016, 7, 12, 1728–1736), we and others have shown that amyloid fibrils are not damaged with the laser power as high as 200 mW. Thus, we conclude that the laser irradiation at the intensities used only induces reversible order-disorder transition in the exposed area. Additionally, in other earlier works from our group on amyloid fibrils and cellulose nanocrystals, we have shown experimentally and numerically the formation of tactoids with timescales of the order of a few minutes (*Nat Commun* 2022, 13, 2778, *Nat Commun* 2023, 14, 607). This behavior of tactoid formation, the symmetry of the nematic field and the structural relaxation of mesogens were successfully explained following thermodynamics principles in these works.

Comment 3: In the Discussion section, the authors concluded the contribution of enthalpic process in the tactoid formation and manipulation against the Onsager theorem assuming entropic process. For my understanding, however, the Onsager theorem does not predict the tactoid formation but only explains the isotropic-nematic transition. Also, many experimental studies have demonstrated that real systems do not rigidly obey the theorem. Contribution of non-entropic, i.e. enthalpic process should not be a new finding. In that sense, the results should be discussed in relation to the formation mechanism of tactoids.

Response: The referee is entirely correct by saying that Onsager theory determines only the boundaries of isotropic and nematic phase, and that tactoids formation is more

properly described by other theories involving elastic energy distortion, first in line being the Frank-Oseen energy functional (see for example our works starting with *Nature Nanotechnology*, 13, 330–336 (2018)). However, we note that in our work the input of energy in the form of NIR irradiation (i.e. heat dissipation, enthalpy) leads of a nematic-to-isotropic transition **in the entire irradiated volume** irrespectively of the tactoid volume or symmetry. Therefore, the order-disorder transition discussed in the manuscript remains of “bulk” nature, where by “bulk” we mean the entire irradiation volume. Therefore, under the conditions used, the Onsager theory remains the reference theory to understand the energetic driving force restoring the liquid crystalline ground state after irradiation. From there, the observation that non-entropic (enthalpic) contributions lead to liquid crystalline melting, as opposed to the Onsager theory findings, is relevant and highlight a non-trivial finding. The observation made by the referee becomes important if the irradiation volume is larger than the single tactoid volume. These experimental conditions are not used in the manuscript, but we take inspiration from this point to design corresponding experiments in our future work.

Comment 4: Molecular mechanism of the tactoid collapse, reformation, and structural modification should be given.

Response: We are thankful to the reviewer for this constructive suggestion. Again, we must note to start that order-disorder-order transitions occur within the full irradiation volume, which makes the problem under the lens a “bulk” study. This being said, if by “molecular” mechanisms, the reviewer refers to “mechanistic”, the melting process upon NIR irradiation is (in the ideal case) an order-disorder process associated with a change in the relative contributions of free volume and rotational entropies, as predicted by the original Onsager theory, with the main difference that the trigger is here the injection of enthalpy via NIR irradiation. As noted on the comment above, this is a surprising finding that we have already commented in the manuscript. The reformation and structural evolution back to the ground LC state from the isotropic phase is, on the contrary, a disorder-order transition which is very well described by the Frank-Oseen theory and already discussed in several earlier publications from us and other groups. For a review on this process, we address the Reviewer to our recent review: *Reports on*

Progress in Physics 88 (3), 036601 (2025).

If, on the other hand, by “molecular” mechanisms, the reviewer refers to “molecular scale”, the amyloid mesogen length scale of ~ 435 nm in length and 4 nm in diameter makes the complexity of the problem intractable at the molecular level. Any study on molecular mechanisms of first order transitions, which we agree is of high importance, deserves a full separate story possibly via coarse-grained MD simulations, as fully atomistic simulations would be extremely time consuming. This is a topic which deserves a fully study on its own.

Reviewer #2

This work describes manipulation of amyloid colloidal liquid crystalline structures, tactoids, by micron scale targeted heating with a laser. The efforts done to visualize these phenomena as images, in short time scale, are very impressive and results highly interesting to learn from. Effects due to manipulation have been demonstrated from multiple angles making it interesting from application point of view. The different responses between bipolar and cholesteric tactoids to manipulation are very well documented and explained for further utilization. It is curious to think that the 3PA technic would work on cellulose nanocrystal (CNC) tactoids as well, but since there's no aromatic functionalities, should the CNC material be somehow functionalized (chemically or with additives) to probe the technic on CNCs? I recommend publishing as it is. Beautiful work!

Response: We are happy that the Reviewer finds our work impressive and beautiful. We thank the Reviewer for his/her encouraging comments. We tried the effect of the 3PA on CNCs tactoids (no functionalities, Supplementary Fig. 10) and observed no visible changes in the tactoids' structure. CNCs tactoids were actually used as control, precisely because free of aromatic groups capable of absorbing radiation. As the Reviewer precisely suggested, functionalizing the CNC would be a next promising step in an attempt to manipulate CNC tactoids with MPA, and we may take inspiration from this comment for our future work.

Supplementary Fig. 10 | Exposure of CNCs tactoids. **a.** The molecular structure of CNCs and the UV-VIS spectrum of CNCs suspension. The 4% CNCs suspension shows the formation of birefringent domains. **b.** AFM images of CNCs (diluted $\times 500$). The inset image is the CNCs film on Mica. The scale bar is 500 nm. **c.** POM image of CNCs cholesteric tactoids after an incubation of 4 days. The red region is prepared for the laser exposure. POM image was captured under a crossed polarizer. The scale bar is 60 μm . **d.** POM image checked immediately after the exposure. No visible changes occur in the cholesteric structure exposed in the marked region. The scale bar is 60 μm .

Reviewer #3

Lin et al. show that amyloid tactoids can be manipulated using a pulsed laser. Heating droplets with high spatial accuracy, they control individual droplet morphology. The authors demonstrated this principle using two types of tactoids: bipolar and cholesteric. The morphological changes, particularly in the cholesteric tactoids, were interesting, as they could manipulate the number of bands. The authors described their process in detail and characterized the internal morphological changes. A particularly notable aspect of the work is the tactoids' ability to self-heal, which opens possibilities for many technological applications as well as fundamental physical studies. The authors also presented an application of their technique using a cholesteric tactoid as an optical switch, demonstrating both fundamental insights and practical applications. The reviewer would like to congratulate the authors on this work and finds it to be of high quality and suitable for the standards of Nature Communications. The reviewer particularly appreciates that this is curiosity-driven research, which led to interesting fundamental discoveries. The high-quality images and videos will certainly engage readers and demonstrate the possibilities of these findings. The work is ready for publication, but the reviewer would like to address minor corrections and clarifications that will increase the reproducibility and clarity of the work.

The reviewer recommends acceptance of this manuscript following these minor revisions.

Response: We are happy that the Reviewer finds our work high quality study and would like to thank them for their positive comments. We addressed all the suggested revisions in the revised manuscript.

Comment 1: The reviewer is particularly interested in the experimental details of the heating protocol. As far as the reviewer understood, the authors are using a scanning laser microscope, but it is not clear from the methods section whether the laser is moving, or the sample is moving. Scanning inherently involves a delay between different locations. The reviewer would like to know how fast this scanning is, which raises the question of whether any delay can be ignored. The authors should comment on whether this delay in heating times could affect the dynamics of tactoid self-healing. For

example, in supplementary movie 5, the zipping and recovery of the tactoid begins at one end and then zips back. It is unclear whether the laser cuts one part first and then the second part, or if it scans back and forth between the two locations. This also applies to the more complex patterns described in Figures 3e and 3j. Additionally, Figure 1f is not clear regarding power density units. The x-axis should be labeled as power per cm^2 , which for a pulsed laser should be on the order of TW/cm^2 or GW/cm^2 given the description of the optical setup. The y-axis units are also difficult to understand—is this related to the scanning rate? From the insets, the reviewer cannot determine at what power the laser induced the isotropic transition.

Response: We are thankful for this comment of the reviewer which allows us to clarify important experimental details. The irradiation we applied is from a point scanning confocal microscopy (Leica SP8 equipped with Mai Tai XF (Spectra-Physics), tunable from 720–950 nm). The microscopy can sequentially scan a single point or small region of interest (ROI) within a sample. The full frame resolution (laser points) was set as 1024×1024 or 512×512 . As a result, ROI scanning provides us possibility to scanning designed patterns. Both the laser beam can be moved orthogonally to the xy plane by high-speed oscillating mirrors and the sample along the z axis by a z piezo stage. The single scan speed is 5-7 full frames per second. Thus, once the designs (for example, the three striped-shaped regions as in Movie 5) are set, the laser irradiation takes less than one second, indicating that delay between patterns is on the order of microseconds. Considering that in all our discussions about zipping and recovery (for example, see Fig. 3c,d,g,h and Fig. 4d-g,i) involve time scales on the order of hundreds of seconds, the delay can be ignored.

The relevant descriptions have been added to the methods section and tracked in yellow color:

“The microscopy can sequentially scan a single point or small region of interest (ROI) within a sample. The full frame resolution (laser points) was set as 1024×1024 or 512×512 . As a result, ROI scanning provides us possibility to scanning designed patterns. Both the laser beam can be moved orthogonally to the xy plane by high-speed oscillating mirrors and the sample along the z axis by a z piezo stage. The single scan

speed is 5-7 full frames per second.”

We thank the Reviewer for pointing out the issues regarding the axes in Fig. 1f. It should be laser power (mW). Power meter was used here to measure the power of laser from objective (25× 0.95NA L Water HCX IRAPO). In addition, we now changed the y axis to the scanning time of each pixel (laser point) from nematic to isotropic. The scanning time was calculated following single pixel dwell time multiplied by the number of repetitions. The result shows that the irradiation of 4.6 and 9.2 mW would not trigger the structure transition with a total pixel dwell time of 40 μ s. The obvious transition was induced when the laser power increased to 13.6 mW (total pixel dwell time of 16 μ s).

Fig. 1. | The LLCPS of β -lactoglobulin (BLG) amyloid fibrils and the multiphoton adsorption of amyloid tactoids. a. Amyloid fibrils with β -sheet-rich structures were

obtained from the incubation of β -lactoglobulin (BLG) protein at pH=2 and 90 °C conditions. Protein structure is available from the Protein Data Bank (PDB ID: 3BLG).

b. The AFM images of amyloid fibrils before (left panel: top-left) and after (left panel: top-right) being shortened into uniform length, and TEM image (left panel: bottom) of original amyloid fibrils. The scale bar is 500 nm. When prepared at a concentration within the Onsager binodal lines, the suspension of short amyloid fibrils shows the formation of a rich variety of tactoids through LLCPS (right panel, image is taken between crossed polarizers). The schematic shows the director field orientation for three different classes of homogenous, bipolar and cholesteric tactoids. The scale bar is 200 μ m. **c.** Schematic showing the exposure of LC cholesteric structure to the pulsed laser. Compared to the ordinary laser focus, the multiphoton absorption can trigger the heating precisely on a small focused spot. This spot can be scanned spatially in xyz -axes. **d.** Open aperture z -scan transmission curve from BLG fibrils solution with theoretical fitting for three-photon absorption (3PA). The sample was exposed with 100 fs pulses at 810 nm wavelength with a pulse repetition rate of 80 MHz. **e.** Linear absorption spectrum of BLG fibrils with background subtracted at one-photon absorption (1PA) mode. The absorption was measured in the range between 200 and 800 nm. **f.** The relationship between the scanning time (pixel dwell time) of laser-induced nematic-isotropic transition and laser power. The scanning time was calculated following single pixel dwell time multiplied by the number of repetitions. Here, a two-dimensional single scanning layer inside of a cholesteric tactoid was exposed under different laser power. The dark region represents the localized disordered state of amyloid fibrils induced by the laser exposure.

Comment 2: Following the previous question, the reviewer would like to know how the authors controlled the thickness of the exposure region depicted in Figure 3a. Is the objective or sample moving up and down? Given that the laser spot should be smaller than 5 μ m, the reviewer wonders what the protocol is for controlling these regions. The reviewer thinks these questions could be easily answered with a short description of the protocol in the methods section, which would give readers a clearer picture of how the experimental setup works.

Response: The scanning of the thickness is based on the z-piezo stage (not the objective). To control these small regions, we use ROI + z-stack protocol to realize the irradiation of required volumes. In our experiments, we fixed distance between each layer in z stack irradiation at 1 μm . As a result, a 5 μm deep region would cover 5 layers of scanning.

We agree with the Reviewer, and, in the revised manuscript, we added following notes to the Methods section to clearly explain our protocol:

“The scanning of the thickness is based on the z-piezo stage. We used ROI + z-stack protocol to realize the irradiation of required volumes where the distance between each layer in z stack irradiation was fixed at 1 μm .”

Comment 3: Can the authors explain how they concluded that the Rayleigh length of the beam is around 3.5 mm given the numerical aperture of the objective and the wavelength used? This Rayleigh length should be in the submicron realm.

Response: Thank you for the comments. We did not use the objective here. The wavelength is 810 nm and the beam waist radius is 30 μm (measured by the knife edge method). In air,

$$Z_R = \frac{\pi\omega_0^2}{\lambda_0}$$

this result in $Z_R=3.5$ mm. We added following notes to the Methods section to clearly explain our protocol:

“The laser was focused onto the sample by a lens to a minimum beam waist radius of 30 μm measured by the knife edge method.”

Comment 4: Could the authors comment on why cutting along the tactoid horizontally, as depicted in figure 4h, did not change the number of bands, given that figures 5b and 5d show cuts in a similar fashion? Is the pitch control a function of the laser power?

Response: The Reviewer is correct. Cutting the tactoid or manipulating the bands is achieved by controlling the laser power. Cutting or manipulating the pitch depends directly on the laser power. A higher laser power (~ 0.2 nJ per pulse) results in breaking the LC order of amyloid fibrils and consequently results in the separation of tactoids, while a lower laser power (~ 0.14 nJ per pulse) induces a relaxation gap that grow a

new band. In the manuscript, we mentioned the laser power in each manipulation.

Comment 5: When describing the optical switch, how did the authors induce or control the tactoid rotation? Do the authors know why the fluorescence is angle-dependent?

Response: The rotation is realized by rotating the laser. In terms of the angle-dependent fluorescence, we believe that fluorescence, as other optical and physical properties of the tactoids has a tensorial anisotropic distribution with respect to the polar coordinates taken from the helical axis of the chiral nematic field.

Comment 6: It is difficult to discern B1 and B2 in Figure 2i showing the cutting of a bipolar tactoid. Supplementary Figure 6 helped considerably, but during the initial reading it was difficult to understand what B1 and B2 represented. The authors could consider using the B1 and B2 images from the supplementary information as insets.

Response: We drew a new cartoon to help the understanding of B1 and B2 in Fig.2i.

Fig. 2 | Manipulation of amyloid tactoids by 3D-Heating. a. Time-lapse images of irradiation of a single bipolar tactoid with marked exposed region (fully covered). The scale bar is $50 \mu\text{m}$. **b.** Aspect ratio and volume of fully exposed bipolar tactoids in (a) plotted vs. exposure time. The gray curves represent the fittings. **c.** Irradiation of half a bipolar tactoid. The scale bar is $50 \mu\text{m}$. **d.** Aspect ratio and volume of half exposed bipolar tactoids in (c) vs. exposure time. **e.** Time-lapse images of a cholesteric tactoid being fully exposed to irradiation. The scale bar is $60 \mu\text{m}$. **f.** Aspect ratio and volume of the cholesteric tactoid in (e). **g.** Exposure of half cholesteric tactoid. The scale bar is $60 \mu\text{m}$. **h.** Dynamics of half pitch measured in the non-exposed region of the tactoid in (g). The color bar represents the value of half pitch. **i.** Break-up of a bipolar tactoid by 3D-Heating. The images show the cutting process of a large bipolar tactoid into two smaller bipolar tactoids, labeled B1 (inside) and B2 (outside). The thickness of the

exposure region $\Delta z = 5 \mu\text{m}$. The scale bar is $30 \mu\text{m}$. **j.** The aspect ratio and volume of the tactoid (B1) during the bipolar cutting. The gray curve represents the fittings. **k.** A cholesteric tactoid (O1) was cut into two smaller cholesteric tactoids along the band (2R) with $\Delta z = 15 \mu\text{m}$. The resulting tactoids are labeled C1 and C2. The tactoid O2 is marked as the control experiment. The scale bar is $30 \mu\text{m}$. **l.** Measurement of half pitch while O1 transforms to C1 and C2 tactoid. All of the images were obtained under a crossed polarizer.

Comment 7: The reviewer found Supplementary Figure 3 very useful, as it provides a clear picture of the different types of tactoids and how the authors are studying them. The reviewer recommends that the authors consider incorporating this figure into the main text.

Response: Thank you for the suggestion. We added schematics of tactoids from Supplementary Figure 3 into Fig.1b now (see Fig. 1b in Comment 1). In addition, the LC-PolScope images of the three classes of tactoids are kept in Supplementary Fig. 3.

Comment 8: Really minor comments: Figure 1e would benefit from arrows pointing to the peak absorption wavelength and the laser excitation wavelength. Figure 2l regarding the pitch of the bands was much more useful than the heat maps.

Response: We added an arrow pointing to the peak absorption wavelengths in the revised Fig. 1e (shown in comment 1). For the UV-Vis absorption spectrum, we added the description of “The absorption was measured in the range between 200 and 800 nm”. The modified figure is exhibited in comment 1.

In Fig. 2h, the heat map was made to show the half pitch distribution of each band. In order to show the half pitch value directly and clearly, we followed the suggestion from the Reviewer and changed the heat map into cross section profiles like Fig. 2l. Please find the Fig.2h in Comment 6.

Writing, reading, erasing, cutting and self-healing individual amyloid tactoids by multiphoton absorption

Review

Lin et al. show that amyloid tactoids can be manipulated using a pulsed laser. Heating droplets with high spatial accuracy, they control individual droplet morphology. The authors demonstrated this principle using two types of tactoids: bipolar and cholesteric. The morphological changes, particularly in the cholesteric tactoids, were interesting, as they could manipulate the number of bands. The authors described their process in detail and characterized the internal morphological changes.

A particularly notable aspect of the work is the tactoids' ability to self-heal, which opens possibilities for many technological applications as well as fundamental physical studies. The authors also presented an application of their technique using a cholesteric tactoid as an optical switch, demonstrating both fundamental insights and practical applications.

The reviewer would like to congratulate the authors on this work and finds it to be of high quality and suitable for the standards of Nature Communications. The reviewer particularly appreciates that this is curiosity-driven research, which led to interesting fundamental discoveries. The high-quality images and videos will certainly engage readers and demonstrate the possibilities of these findings. The work is ready for publication, but the reviewer would like to address minor corrections and clarifications that will increase the reproducibility and clarity of the work.

1. The reviewer is particularly interested in the experimental details of the heating protocol. As far as the reviewer understood, the authors are using a scanning laser microscope, but it is not clear from the methods section whether the laser is moving, or the sample is moving. Scanning inherently involves a delay between different locations. The reviewer would like to know how fast this scanning is, which raises the question of whether any delay can be ignored. The authors should comment on whether this delay in heating times could affect the dynamics of tactoid self-healing. For example, in supplementary movie 5, the zipping and recovery of the tactoid begins at one end and then zips back. It is unclear whether the laser cuts one part first and then the second part, or if it scans back and forth between the two locations. This also applies to the more complex patterns described in Figures 3e and 3j. Additionally, Figure 1f is not clear regarding power density units. The x-axis should be labeled as power per cm^2 , which for a pulsed laser should be on the order of TW/cm^2 or GW/cm^2 given the description of the optical setup. The y-axis units are also difficult to understand—is this related to the scanning rate? From the insets, the reviewer cannot determine at what power the laser induced the isotropic transition.
2. Following the previous question, the reviewer would like to know how the authors controlled the thickness of the exposure region depicted in Figure 3a. Is the objective or sample moving up and down? Given that the laser spot should be

smaller than 5 μm , the reviewer wonders what the protocol is for controlling these regions. The reviewer thinks these questions could be easily answered with a short description of the protocol in the methods section, which would give readers a clearer picture of how the experimental setup works.

3. Can the authors explain how they concluded that the Rayleigh length of the beam is around 3.5 mm given the numerical aperture of the objective and the wavelength used? This Rayleigh length should be in the submicron realm.
4. Could the authors comment on why cutting along the tactoid horizontally, as depicted in figure 4h, did not change the number of bands, given that figures 5b and 5d show cuts in a similar fashion? Is the pitch control a function of the laser power?
5. When describing the optical switch, how did the authors induce or control the tactoid rotation? Do the authors know why the fluorescence is angle-dependent?
6. It is difficult to discern B1 and B2 in Figure 2i showing the cutting of a bipolar tactoid. Supplementary Figure 6 helped considerably, but during the initial reading it was difficult to understand what B1 and B2 represented. The authors could consider using the B1 and B2 images from the supplementary information as insets.
7. The reviewer found Supplementary Figure 3 very useful, as it provides a clear picture of the different types of tactoids and how the authors are studying them. The reviewer recommends that the authors consider incorporating this figure into the main text.
8. Really minor comments: Figure 1e would benefit from arrows pointing to the peak absorption wavelength and the laser excitation wavelength. Figure 2l regarding the pitch of the bands was much more useful than the heat maps.

The reviewer recommends acceptance of this manuscript following these minor revisions.